# Propensity Score Analysis with Partially Observed Baseline Covariates: A Practical Comparison of Methods for Handling Missing Data

**DOI:** 10.3390/ijerph18136694

**Published:** 2021-06-22

**Authors:** Daniele Bottigliengo, Giulia Lorenzoni, Honoria Ocagli, Matteo Martinato, Paola Berchialla, Dario Gregori

**Affiliations:** 1Unit of Biostatistics, Epidemiology and Public Health, Department of Cardiac, Thoracic, Vascular Sciences and Public Health, University of Padova, 35122 Padova, Italy; daniele.bottigliengo@phd.unipd.it (D.B.); giulia.lorenzoni@unipd.it (G.L.); honoria.ocagli@unipd.it (H.O.); matteo.martinato@unipd.it (M.M.); 2Department of Clinical and Biological Sciences, University of Torino, 10124 Torino, Italy; paola.berchialla@unito.it

**Keywords:** propensity score, missing data, non-interventional studies

## Abstract

(1) Background: Propensity score methods gained popularity in non-interventional clinical studies. As it may often occur in observational datasets, some values in baseline covariates are missing for some patients. The present study aims to compare the performances of popular statistical methods to deal with missing data in propensity score analysis. (2) Methods: Methods that account for missing data during the estimation process and methods based on the imputation of missing values, such as multiple imputations, were considered. The methods were applied on the dataset of an ongoing prospective registry for the treatment of unprotected left main coronary artery disease. The performances were assessed in terms of the overall balance of baseline covariates. (3) Results: Methods that explicitly deal with missing data were superior to classical complete case analysis. The best balance was observed when propensity scores were estimated with a method that accounts for missing data using a stochastic approximation of the expectation-maximization algorithm. (4) Conclusions: If missing at random mechanism is plausible, methods that use missing data to estimate propensity score or impute them should be preferred. Sensitivity analyses are encouraged to evaluate the implications methods used to handle missing data and estimate propensity score.

## 1. Introduction

Non-interventional (or non-randomized) studies are increasingly being used to infer causal relationships between new treatments and health outcomes in real clinical settings [1]. In non-randomized clinical studies, the allocation of subjects into treatment groups often depends on their characteristics collected at baseline. Consequently, the groups of individuals systematically differ from each other in terms of baseline covariates, leading to the so-called confounding bias. In such situations, a naïve comparison of the outcomes between the treatment and control groups would provide a biased treatment effect estimate. Thus, statistical methods that explicitly account for the presence of confounding are needed to make a causal assessment of the effect of a new treatment or therapy.

Propensity Score (PS) methods have been proposed to reduce confounding bias in non-interventional studies and to provide consistent treatment effect estimates [2]. Briefly, PS is defined as the individual probability of being treated given baseline characteristics. As demonstrated by Rosenbaum and Rubin in their seminal paper [3], the PS acts as a balancing score, i.e., the treatment groups are on average balanced in terms of baseline covariates after conditioning on the PS. Thus, conditioning on PS allows the researcher to mimic the settings of an RCT. PS techniques are increasingly becoming popular in non-interventional clinical studies, especially in the cardiovascular and surgical literature [4,5,6,7]. For example, PS methods have recently been used to compare coronary artery bypass grafting (CABG) and percutaneous coronary intervention (PCI) [6,8]. The most popular PS based methods used to minimize confounding bias are Propensity Score Matching (PSM), stratification on the PS, Propensity Score as Inverse Probability of Treatment Weighting (PS-IPTW), and covariate adjustment using the PS [9]. Recently, Full-Matching (FM) on PS, which can be thought of as a synthesis of stratification and PS-IPTW, is gaining attention in the applied literature [10,11].

The estimation of PS requires the individual baseline characteristics to be fully observed. In practice, however, some values in baseline covariates are missing. When missing data arises, individual PS cannot be estimated for individuals with one or more missing values in baseline variables. The standard approaches to deal with missing data in PS analysis are complete case (CC) analysis and the missing indicator method (MIND) [12,13]. Popular alternatives to classical techniques are represented by methods that impute missing data, such as Multiple Imputation (MI) [14], and statistical models that include the missingness during the estimation of PS, such as Generalized Boosted Modeling (GBM) [15]. To the best of our knowledge, there is a paucity of literature on which an approach for dealing with missing data provides more advantages from a practical point of view, especially when assumptions on the missingness mechanism cannot be assessed using observed data. Only a few studies performed a comparison of different strategies for handling missing in PS analysis [16,17,18].

The present study aims to compare several missing data strategies in combination with different PS based methods. We used as a motivating example a dataset from an ongoing multicenter prospective registry of the treatment of unprotected left main coronary artery disease. Since the registry data collection is still ongoing, the results and findings of the study have not yet been published. Thus, we will focus only on the design stage of the PS analysis. That is, we are interested in evaluating the balance of baseline covariates obtained with the examined statistical methods rather than estimating a causal quantity of interest. The remainder of this paper is organized as follows. In Section 1.1, we describe the data of the motivating example. In Section 1.2, we introduced the PS methodology. In Section 2, we introduce the PS and missing data framework, describe the methods used in the study, and the implementation of the statistical analysis. In Section 3, we present the results of the analysis. In Section 4, we discuss the findings and contextualize them with the current literature. In Section 5, we provide some practical recommendations for the implementation of the methods.

### 1.1. Motivating Example

The main objective of the registry was to evaluate the safety and efficacy of a new generation stent in the treatment of unprotected left main coronary artery disease (ULMCAD), both isolated or in association with two- or three-vessel coronary artery disease. The registry has been designed to be comparable in terms of primary endpoint to the PCI arm of the Evaluation of XIENCE Everolimus-Eluting Stent Versus Coronary Artery Bypass Surgery for Effectiveness of Left Main Revascularization (EXCEL) and Nordic-Baltic-British Left Main Revascularization Study (NOBLE) trials [19,20]. Study details are not disclosed as per the research agreement.

A subgroup of enrolled patients was expected to be treated with intravascular imaging-intravascular ultrasound (IVUS) or optical coherence tomography (OCT) during PCI, which are intracoronary imaging techniques that aid clinicians in optimizing stent implantation and that were shown to improve patient’s prognosis compared with standard angiography-guided PCI [21,22,23]. We focused on the pre-specified analysis of the study that aims to evaluate the impact of the type of imaging on the clinical outcomes by comparing IVUS, OCT, and standard angiography-guided groups of subjects.

At the onset of the study, data on 531 patients enrolled in 26 centers were available at the investigators, with information on demographic, clinical, and procedural characteristics. Data collection was managed using REDCap electronic data capture tools [24,25]. Among the enrolled patients, 263 underwent standard angiography-guided, 200 underwent IVUS, 15 underwent OCT, and 16 underwent fractional flow reserve (FFR), whereas for the remaining subjects the imaging technique was not assessed yet. The illustration of the methodology for estimating PS with missing data and assessing the balance of baseline covariates was performed by comparing standard angiography-guided groups with a group of subjects that underwent both IVUS and OCT. A separate comparison between IVUS and OCT was not considered since the small number of patients in the OCT groups would make the comparison inaccurate. Thus, the final dataset included 478 patients (263 in the standard angiography-guided group and 215 in the IVUS-OCT group).

### 1.2. Propensity Score Framework

The PS is defined as the individual probability that a subject receives the treatment given the observed values of the pre-treatment covariates. Denoting with *i = 1, …, n* the *i-th* subject among the *n* subjects enrolled in the study, with *T_i_ = {0,1}* the treatment received by the subject, and with *x_i_* the observed values of the pre-treatment covariates, the PS can be formalized as:π (x_i_) = P(T_i_ = 1|X_i_ = xi)(1)

Rosenbaum and Rubin [3] showed that for a specific PS value the average difference of the outcomes between treatment and control groups provides an unbiased estimate of the treatment effect if the following conditions hold: (i) no unmeasured confounding, and (ii) every subject has a non-zero probability of receiving the treatment (positivity assumption).

PS estimation is often conducted using classical logistic regression (LR). Multiple methods to estimate PS have been described, such as MLTs, which are potentially able to flexibly estimate treatment assignment mechanism, and covariate balancing propensity score (CBPS), which aims to estimate PS model by maximizing the balance of pre-treatment covariates [26,27,28].

Common techniques to estimate treatment effects using PS include PSM, PS-IPTW, stratification on the PS, and covariate adjustment using the PS. PSM consists of forming a matched set of treated and control observations with similar PS values. The most common implementation of PSM is 1:1 nearest neighbor (NN) matching, which consists of pairing one treated and one control unit that share close estimated PS. A caliper is often imposed to improve the quality of the matched set. PS-IPTW technique creates weights based on the estimated PS such that a new artificial sample of units, balanced in terms of measured baseline covariates, is obtained. Individual weights are computed as:w_i_ = T_i_/π_i_ + (1 − T_i_)/(1 − π_i_)(2)

Weights created using the PS-IPTW method are conceptually like those used in complex survey design studies to make the sample at hand representative of the target population. Thus, the weights are used to compute the summary statistics of the sample, e.g., measures used to evaluate the balance, or difference in the outcomes means. Stratification on the PS partitions the observations into groups based on PS values, e.g., quintiles of the estimated PS. Individuals in the same stratum will have similar PS values, which should result in balance in terms of covariates between treatment and control groups in each stratum. Once strata are defined, the treatment effect estimate on the whole sample can be obtained by thinking of each stratum as a single RCT and pooling stratum-specific estimates using meta-analysis methods. Stratification on PS has been shown to remove 90% of the bias in the treatment effect estimate [13]. Covariate adjustment using PS consists of fitting a regression model using as covariates the treatment indicator and the estimated PS. The treatment effect is then represented by the coefficient of the regression model associated with the treatment variable. Previous studies have shown that this is generally not a statistically valid method [29,30,31]. FM is a PS based method that can be thought of as a mix of stratification on PS and PS-IPTW. In summary, units are partitioned into strata with at least one treated and one control such that the within-strata difference in PS between treated and controls is minimized. Once strata are obtained, different sets of weights can be constructed to obtain the treatment effect estimate of interest [32]. The main advantages of FM over other PS based methods are: (i) it retains all the units in the final sample, avoiding the risk of bias when discarding observations from the initial set of individuals; (ii) FM permits the estimation of different causal effects based on the type of weighting scheme that is used [33].

## 2. Materials and Methods

### 2.1. Propensity Score and Missing Data in Non-Interventional Studies

One of the major issues in PS analysis occurs in the case of missing data in baseline covariates. Assuming that the treatment indicator is fully observed, two methods are often advocated to handle missingness in PS analysis: CC and MIND. CC consists of estimating the PS only for those individuals with observed values for all the covariates included in the treatment assignment model. CC is known to provide unbiased treatment effect estimate if the missingness mechanism is Missing Completely At Random (MCAR), i.e., the missingness depends neither on observed and unobserved data, although statistical power is reduced due to the loss of observations. When the mechanism that generated missing data is Missing At Random (MAR), i.e., the missingness depends only on observed variables, or Missing Not At Random (MNAR), i.e., missing data are generated given information not observed in the dataset, there may be the risk of obtaining a biased estimate of the treatment effect [34]. The MIND [13] approach allows estimating PS for each unit, even if some covariate values are not observed. Briefly, missing values are replaced with an arbitrary specific value. When the covariate is continuous, the value 0 is often used to replace the missing value, whereas a “missing” category is used for categorical covariates. Furthermore, an additional indicator variable that assumes value 0 if the corresponding value is observed and 1 otherwise is created. The missing indicator variables are then used as covariates in the PS model. The MIND approach was shown to be equivalent to the Missing Pattern Approach (MPA) proposed by Rosenbaum and Rubin [13] with a single partially observed covariate. Additionally, it represents a simplified version of MPA in more complex scenarios, e.g., one fully observed covariate and one partially observed covariate, with the additional assumption that the association between the fully observed covariate and the treatment is the same even if the other covariate is partially observed [35]. The MIND approach was shown to introduce bias in regression models [36,37], and its role has been questioned in PS analysis [38].

Alternative methods that handle missing data during the fitting process are represented by PS estimation with GBM and logistic regression with a stochastic approximation of the EM algorithm (SAEM). GBM is an MLT that allows a flexible estimation of PS by modeling nonlinearities, interactions, and including a large number of potential confounders [27]. PSs are estimated by an iterative process that uses regression trees to model the treatment assignment mechanism. Moreover, the parameters of the GBM algorithm can be tuned such that the balance between treatment and control groups in terms of baseline covariates is maximized [39,40]. The use of GBM in non-interventional studies has been increasing recently [41,42,43,44]. The main advantage of GBM in PS analysis with partially observed covariates relies on how GBM deals with missing data during the estimation process. That is, each regression tree built by GBM handles observations with missing data using surrogate splitting, i.e., a separate node for the missingness is created by the algorithm. PSs are then estimated even if some covariates are partially observed. The SAEM allows for the estimation of PS using logistic regression with a method that explicitly deals with missingness in the covariates [45]. The logistic regression is fitted with a stochastic approximate version of the EM algorithm, which is based on the Metropolis-Hastings algorithm. The algorithm makes use of a fitting process that stochastically computes the conditional-expectation of the likelihood function based on the fully observed data without any needs to approximate it with a large number of Monte Carlo simulations. The SAEM method assumes that the mechanism that generates missingness is MCAR or MAR. The use of the algorithm in PS analysis has been advocated by the authors as an appealing approach to deal with missing data. Recently, the estimation of PS with SAEM method was explored by a study from Mayer and colleagues who aimed at evaluating the performances of several causal inference methods with missing data [46]. The authors found that the approach had good performances across a wide range of simulated scenarios. However, they noticed that treatment effect estimates may have some bias if the underlying assumptions of the method are not fully supported by the data.

MI is a well-established method to handle missing data in medical studies [47,48,49]. Previous studies found that MI outperforms many alternative methods for handling missingness in covariates [50,51,52]. The classical MI approach is implemented in the following steps: (i) a model for the missing data is specified; (ii) missing values are imputed *m* times by sampling from the posterior distribution of the parameters of the missing data model; (iii) *m* complete datasets are generated, and the statistical analysis is performed separately in each dataset; and (iv) the results of the *m* analyses are pooled using Rubin’s rule [14]. When using MI to deal with missing data in PS analysis, two main approaches have been proposed: the “within” strategy and “between” strategy [53]. Both approaches multiply impute missing data and then estimate individual PS in each dataset. After that, the “within” approach estimates the treatment effect in each dataset and pooled the estimates using Rubin’s rule to perform inference, whereas the “between” approach averages the individual PS estimated in each complete dataset and performs inference using the pooled PS values. Multiple manuscripts compared the performances of the “within” and “between” approaches using simulation studies [52,54,55,56]. The results from previous studies did not agree on which strategy should be used in practice. However, a recent work by Granger and colleagues found that the “within” approach should be preferred since it produces less biased estimates and it accounts for between-imputation variability [57].

### 2.2. Statistical Analysis

#### 2.2.1. Propensity Score Estimation

We considered the following models to estimate individual PS: LR, CBPS, and GBM. The following baseline characteristics were included in the PS models: gender, age, body mass index (BMI), hypertension, diabetes, smoker (non-smoker vs. previous smoker or current smoker), coronary artery disease (CAD), hyperlipidemia, previous PCI, previous myocardial infarction (MI), previous stroke/transient ischemic attack (TIA), chronic obstructive pulmonary disease (COPD), peripheral artery disease (PAD), clinical presentation (NSTEMI, STEMI, stable CAD, unstable angina, and other), New York Heart Association (NYHA) classification, Canadian Cardiovascular Society (CCS) classification, EuroSCORE II, glomerular filtration rate (GFR), hemoglobin, left ventricular ejection fraction (LVEF), aspirin, thienopyridine, Syntax Score, lesion to left anterior descending coronary artery (LAD), lesion to left circumflex coronary artery (LCx), and lesion to right coronary artery (RCA). When PS was estimated with LR and CBPS, the baseline covariates were included in the model as main effects. When PS was estimated with GBM, the default parameter values of the model as implemented in the twang R package [58] were used. Three PS based methods were considered: NN matching, FM, and PS-IPTW. NN was implemented as the classical 1:1 matching without replacement. A caliper equal to 0.2 of the standard deviation of the logit of PS was imposed to improve the quality of the resulting matched set [59].

#### 2.2.2. Missing Data Methods

In addition to the traditional methods used to handle missing data in PS analysis, i.e., CC and MIND, we considered the following techniques that include the missingness during the estimation procedure: GBM with surrogate splitting and SAEM. GBM with surrogate splitting was implemented using the twang R package (version 1.6) [58] with default parameter settings. The *misaem* R package (version 1.0.0) [60] was used to implement PS estimation with SAEM method. Furthermore, we considered three different methods based on MI to impute missing data before PS estimation: MI with the fully conditional specification (FCS) [61] method via chained equation algorithm as implemented in the mice R package (version 3.10.0) [62,63], MI with FCS method that combines sequential regression and Bayesian Bootstrap Predictive Mean Matching (BBPMM) as implemented in the *BaBooN* R package (version 0.2-0) [64] via chained algorithm, and MI with the FCS method with additive regression models, bootstrap, and Predictive Mean Matching (PMM) as implemented in *aregImpute* function of the *Hmisc* R package (version 4.4-1) [65,66]. These three methods were labeled as MI-MICE, MI-BBPMM, and MI-AREGIMP, respectively. The MI-MICE method was implemented using PMM for continuous variables, logistic regression for binary variables, and multinomial logistic regression for categorical variables. Twenty imputed datasets were generated after 20 iterations. We did not choose the number of imputed datasets using the fraction of missing data (FMI) as previously suggested [67] since we focused on the design stage of the analysis without any knowledge of the outcomes. The MI-BBPMM method was implemented with the default settings of the *BaBooN* package, generating 20 imputed datasets after 20 iterations. The default settings of the *aregImpute* function were used to implement the MI-AREGIMP method. Twenty imputed datasets were generated after discarding the first 20 iterations of the algorithm as “burn-in”. Regarding the MI methods, for each imputed variable, all the remaining baseline characteristics and the treatment indicator were used as covariates in the imputation model. We implemented the design stage of PS analysis with MI using the “within” approach. We assessed the similarity between the PSs estimated by each method using the Intraclass Correlation Coefficient (ICC), a measure of agreement between continuous measurements. We used the average PS values obtained in each imputed dataset when MI methods were considered.

The structure of missing data in the dataset was explored using an approach based on decision tree, as suggested by Tierney and colleagues [68]. The method was implemented by fitting a regression tree on the data using as dependent variable the proportion of missing values. The fitting process was performed using the *rpart* R package (version 4.1-15) [69], with default settings.

#### 2.2.3. Measures of Balance

Covariate balance of the sets of individuals obtained after PS methods was assessed using the standardized mean difference (SMD) of the PS distributions in the two groups of patients, the overlapping coefficient (OVL), and the C-statistic [70]. The OVL is the proportion of overlap between the density functions of the PS distributions. It ranges between 0 and 1, and higher values represent higher degree of overlap. We considered 1-OVL to make it comparable with other measures of balance. The C-statistic is the area under the receiver operating characteristic (ROC) of PS in the sample of individuals reconstructed after the application of PS methods. It ranges between 0.5 and 1, and lower values indicate better balance. We computed 1-C-statistic to perform a fair comparison with other balance statistics. When the balance was assessed on the imputed datasets generated with MI, we computed the median, the minimum, and the maximum of the measures of balance to account for the variation induced by the imputation process. We considered measures that evaluated the overall balance of the final dataset by targeting the discrepancies in the PS distributions after PS methods were implemented. The choice was motivated by the practical need to simultaneously evaluate several methods and to provide an intuitive and easy-to-read comparison of the achieved balance. We provided more details on the goodness of the sets obtained after PS analysis by computing the proportion of retained individuals (PRI), i.e., the proportion of patients that were not discarded from the final set. When NN was used, the PRI was equal to the number of patients in the matched set divided by the original number of patients. For FM, the PRI was always equal to 1 since FM creates a matched set including all the observations from the original sample. When PS-IPTW, we computed the PRI using the effective sample size as proposed by Kish [71], i.e., the sample size of an unweighted set of observations with approximately the same precision as the weighted sample that was under consideration.

The statistical analysis was implemented using R software for statistical computing (version 4.0.2) (R foundation for statistical computing. Vienna, Austria) [72]. The *MatchIt* R package (version 3.0.2) (R foundation for statistical computing. Vienna, Austria) [73] was used for NN matching and FM, whereas PS-IPTW was implemented using *WeightIt* R package (version 0.10.2) (R foundation for statistical computing. Vienna, Austria) [74]. PSs with CBPS were estimated using the *CBPS* R package (version 0.21) [75]. The R code to reproduce the analysis workflow on a simulated dataset is available on Github (https://github.com/UBESP-DCTV/ps.missing (accessed on 28 May 2021)).

## 3. Results

Table 1 shows the distributions of baseline covariates stratified by imaging group, i.e., angiography and IVUS/OCT. Continuous variables are expressed as 1st quartile, median, and 3rd quartile, and percentages (total numbers) are used for categorical variables. The SMDs were reported to evaluate the degree of imbalance separately for each covariate. SMDs higher than 0.25 suggest a considerable amount of imbalance in the distributions of the baseline covariates between the two groups [76,77]. The most notable differences between angiography-guidance and IVUS/OCT were observed in terms of the proportion of subjects with NSTEMI (higher in the angiography group), the EuroSCORE II (lower in the IVUS/OCT group), the proportion of patients with an FE between 30–50%, which was lower in the IVUS/OCT group, the Syntax score, higher for patients who underwent angiography-guidance. Furthermore, LCx and RCA were less frequently observed in the IVUS/OCT group. Overall, patients that underwent angiography-guidance had more severe clinical conditions on average.

### 3.1. Missing Data

Overall, the percentage of missing data in the sample was 6.3%. In Figure 1 the percentage of missing data for each baseline variable is depicted. The Syntax score and the EuroSCORE II were the covariates that presented the highest percentages of missingness, more than 60% and almost 40%, respectively. The CSS classification and BMI resulted had between 10% and 20% of missing values, whereas percentages below 10% were observed for the remaining baseline characteristics.

Figure 2 shows the decision tree that was used to explore the structure of missingness. The CCS classification was identified as the variable most associated with the proportion of missingness (root node of the tree). Patients who took Thienidopiridine and that were labeled in the third and fourth levels of CCS classification had the lowest proportion of missingness. In contrast to that, the highest proportion of missingness was observed for patients with the following characteristics: CCS equal to 0, 1, or 2, did not take Thienidopiridine, had NYHA class equal to 3 or 4, hemoglobin lower than 13 g/dL, and were less than 71 years old.

### 3.2. Propensity Score Estimation and Common Support

An ICC equal to 0.73 (95% CI; 0.70–0.75) suggests a moderate overall agreement between the estimated PSs [78]. The pairwise correlations, computed with Pearson’s coefficient, between the approaches are graphically depicted in Figure 3. Overall, the correlation coefficients were always higher than 0.5, suggesting a moderate correlation between PSs estimated with different strategies. SAEM method, and methods based on MI with PS estimation performed using LR or CBPS showed the highest correlation, with correlation coefficients close to 1.

Table 2 reports the measure of overall balance in the set of patients before the application of PS methods to evaluate the common support of angiography-guidance and IVUS/OCT groups. When PSs were estimated using GBM, the highest degree of dissimilarity in terms of PS distributions was observed in all scenarios. Good common support was observed when PSs were estimated using LR or CBPS.

### 3.3. Balance

Table 3 reports the measure of overall balance in the set of patients after the application of PS methods to balance the dataset. The datasets that resulted in the highest imbalance were obtained when classical methods to deal with missing data were used, i.e., CC and MIND. Regarding PS based methods, NN and FM provided the most balanced sets of patients in most of the situations. When MI was performed, the overall balance was often satisfactory, with median SMD, OVL, and C-statistic values generally below 0.1, except when PSs were estimated with GBM and PS-IPTW was applied. The highest degree of balance was observed when SAEM and GBM (with surrogate splitting) were used to handle missing data. When NN matching was done, the overall balance was always perfect, with a balance statistics value equal to 0. Similarly, a good balance was achieved when FM was used (values always lower than 0.1 for SMD, OVL, and C-statistic). However, the imbalance was apparent for GBM (surrogate splitting) when PS-IPTW was applied, whereas satisfactorily balanced sets of individuals were obtained with SAEM and PS-IPTW. When NN was used, the PRI was often lower than PS-IPTW. As expected, FM retained all the patients in the final balanced dataset, and it may guarantee higher generalizability of the treatment effect estimates compared to NN and PS-IPTW.

## 4. Discussion

The findings of our study have practical implications. Regarding missing data, several guidelines in the literature have been proposed on the percentage of missingness above which observations should not be discarded. When 5% of missingness arises, the benefits provided by methods such as MI have been suggested to be negligible [79]. Moreover, if more than 10% of the missingness arises, bias is likely to occur when observations with missingness are discarded [34]. We observed an overall percentage of missingness of above 6%. However, some important prognostic factors showed higher percentages of missingness. Simply discarding observations with at least one missing value would results in a considerable loss of information and would enhance the risk of introducing bias in case of MAR mechanism. In such situations, methods that account for missing data should be considered.

In terms of overall balance, we found that classical methods that handle missing data in PS analysis, i.e., CC and MIND, have the worst performances. Higher values of SMDs, OVLs, and C-statistics suggest that some residual imbalance may be present in the final sets of patients, and some adjustments in the outcome analysis should be considered to mitigate the eventual presence of confounding bias. The imbalance was even higher when PS-IPTW was applied, especially when PSs were estimated with GBM. The highest balance was observed when methods that account for missing data during the estimation process, i.e., SAEM and GBM (with surrogate splitting), were considered. When NN matching and FM were applied, the final sets resulted in the largest balance, with SMD, OVL, and C-statistic values often below 0.1. However, the same pattern was not observed for PS-IPTW, which showed results similar to CC and MIND. The results obtained with the SAEM method are promising. The use of such an approach for PS estimation deserves further exploration to assess if it can provide some benefits, as also suggested by the authors [45]. However, the underlying assumptions of PS estimation with SAEM, i.e., the MAR mechanism is plausible, should be carefully explored before using the method. A recent study [18] found that PSs estimated GBM with surrogate splitting returned biased treatment effect estimates, which is somewhat in contrast with the findings of our study that assessed an overall good balance when the method is applied. This may represent an area for future research. Methods based on MI showed competitive performances to SAEM and GBM with surrogate splitting, with average values of SMD, OVL, and C-statistic often below 0.1. Moreover, the maximum values of balance statistics computed on the imputed datasets were below 0.2 in most of the situations, suggesting that balance was achieved many times across all the imputations. The worst overall balance was observed when PSs were estimated with GBM, and PS-IPTW was considered. The choice of which method should be preferred between MI and classical approaches such as CC and MIND is not clear. Previous studies found that MI overcomes classical CC and MIND performances [56,80]. However, a recent study suggested that although some statistical efficiency is lost, CC and MIND may be preferred to MI since they can be used in synergy to control for unmeasured confounding [81]. The results of our study are in line with those that found a superiority of MI to CC. Indeed, the topic should be further explored to understand the performances of the methods in different settings.

Regarding the method used to estimate PS, we found that when GBM was used, PS distributions between angiography-guidance and IVUS/OCT patients were more different than when LR or CBPS were used. Therefore, the sets of patients obtained after the application of PS methods still showed a considerable amount of residual imbalance, especially when PS-IPTW was considered. This issue may be related to the estimation of more extreme PSs values obtained with GBM, which could undermine the positivity assumption. These findings agree with the study of Alam and colleagues [82] who observed better covariate balance and lower bias when PS was estimated with LR or Super Learner than GBM.

PS methods based on matching, i.e., NN matching and FM, resulted in a better balance of the overall set of patients than PS-IPTW. Furthermore, FM did not discard any subjects from the final sets. This feature of FM may represent an advantage since it guarantees higher generalizability of the subsequent estimates than classical NN matching and PS-IPTW. Moreover, FM allows the estimation of different causal effects using different types of weighting schemes.

Our study has some limitations. First, the methods were compared only in terms of the balance of the sets of patients. The choice was motivated by the fact that the registry is still ongoing, and the follow-up is not yet completed. Thus, we focused only on the design stage of the analysis without any knowledge of the clinical endpoints. Once the follow-up will be terminated, further research will be performed by implementing simulation studies that compare the methods in terms of bias and coverage of the treatment effect estimates. Based on the results of the present study, we would expect to observe the least biased estimates when PS is estimated with SAEM and NN or FM are applied. We would expect good performances when missing data are imputed with MI methods and NN or FM are used after PS estimation via CBPS. Besides, it would be interesting to evaluate the coverage of the methods when estimators for time-to-event outcomes are applied, as the primary endpoint of the study. Second, we considered methods for handling missing data that rely on the assumption that the missingness mechanism is MCAR or MAR. Although such an assumption is often stated in practice, deviations are often impossible to test from observed. Future research studies should consider methods that assume an MNAR mechanism and examine how the proposed methods perform in MNAR settings.

## 5. Conclusions

In summary, when missing data arise in PS analysis, the mechanism of missingness in the dataset should be carefully studied to understand which are the factors that affected the missingness itself. If MCAR o MAR mechanisms are plausible, we found that methods that explicitly account for missing data provide better results than classical CC or MIND, especially when the proportion of missingness is important for prognostic factors. We recommended the estimation of PS with the SAEM method and applying FM to balance the groups of patients in terms of baseline covariates. However, we suggested evaluating the plausibility of the SAEM assumptions, since the method can introduce some bias in the treatment effect estimate if the missingness mechanism deviates from MAR. Furthermore, we encouraged the researchers to perform sensitivity analyses by implementing different strategies to handle missing data and to estimate PS and comparing them using measures of overall balance.

## Figures and Tables

**Figure 1 ijerph-18-06694-f001:**
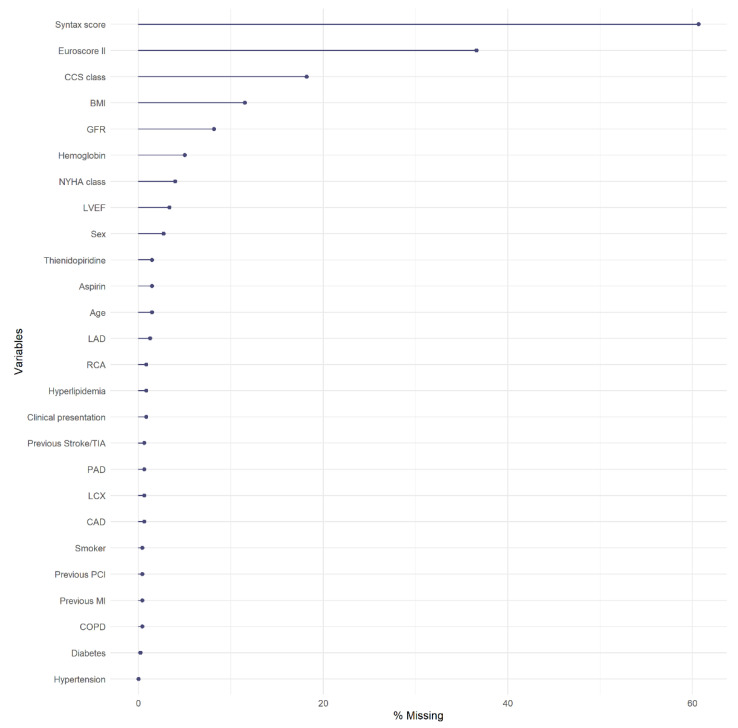
Percentages of missing data for each baseline covariates in the original sample of patients.

**Figure 2 ijerph-18-06694-f002:**
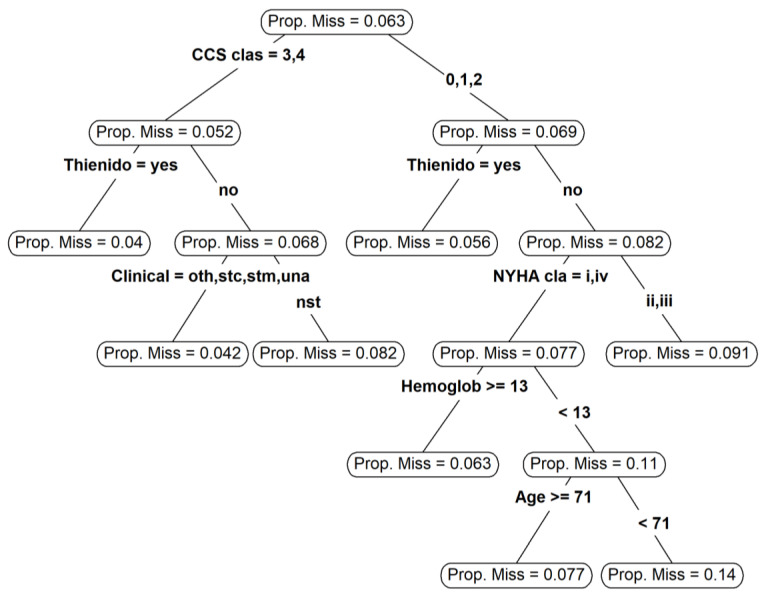
Plot of the decision tree used to explore the structure of missing data. Ellipses represent the nodes of the tree. For each node, the proportion of missingness for the observations is shown. The arrows are the branches of the tree and the variable that determined the split is shown for each node and subsequent branches.

**Figure 3 ijerph-18-06694-f003:**
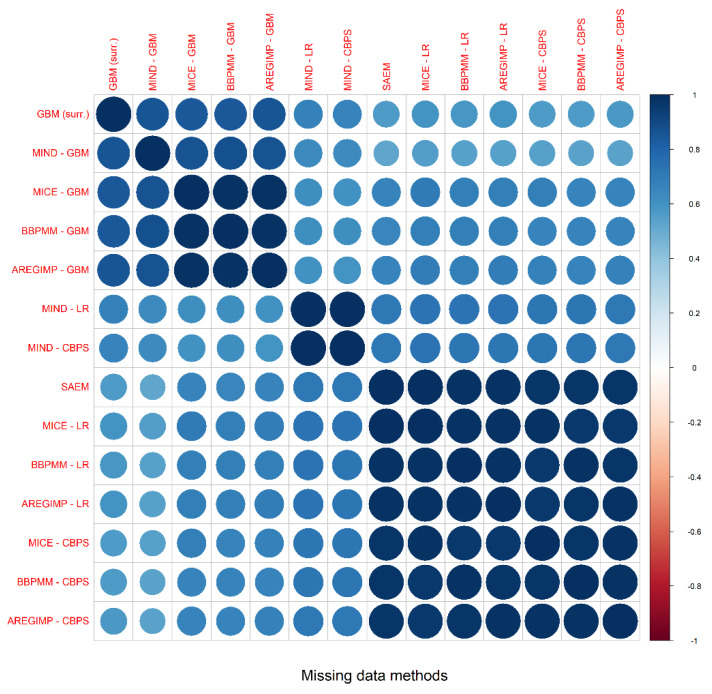
Correlogram of the PS estimated with methods that explicitly account for missing data. The values correspond to the Pearson’s correlation coefficients computed for each pair of methods. The color represents the magnitude of the correlation. The acronyms in red represent the labels of the missing data methods.

**Table 1 ijerph-18-06694-t001:** Descriptive statistics of the sample of patients stratified by angiography-guidance and IVUS/OCT groups. Continuous variables are represented with I quartile/median/III quartile and categorical variable with percentage (absolute numbers). The Standardized Mean Differences (SMDs) on the unbalanced sample are reported in the last column of the table.

Variable	N	Combined (N = 478)	Angiography Guidance (N = 263)	IVUS or OCT (N = 215)	SMD
Gender: male	465	83% (385)	80% (205)	86% (180)	0.14
Age	471	65/72/79	65/73/80	64/72/79	−0.06
BMI	423	24/26/28	24/27/29	24/26/28	−0.01
Hypertension	478	78% (371)	79% (207)	76% (164)	−0.06
Diabetes	477	30% (142)	30% (79)	29% (63)	−0.02
Smoker	476	46% (221)	45% (119)	48% (102)	0.04
CAD	475	25% (117)	21% (56)	29% (61)	0.17
Hyperlipidemia	474	66% (311)	62% (162)	70% (149)	0.18
Previous PCI	476	35% (167)	34% (88)	37% (79)	0.06
Previous MI	476	25% (117)	25% (65)	24% (52)	−0.01
Previous stroke/TIA	475	5% (25)	5% (13)	6% (12)	0.03
COPD	476	7% (34)	7% (18)	7% (16)	0.02
PAD	475	15% (73)	16% (42)	14% (31)	−0.05
Clinical Presentation: NSTEMI	474	31% (145)	37% (96)	23% (49)	−0.30
Other		11% (54)	10% (25)	14% (29)	0.13
Stable CAD		33% (155)	31% (81)	35% (74)	0.08
STEMI		11% (50)	10% (27)	11% (23)	0.01
Unstable Angina		15% (70)	12% (32)	18% (38)	0.16
NYHA: I	459	56% (256)	54% (136)	58% (120)	0.09
II		28% (130)	28% (72)	28% (58)	−0.01
III		13% (59)	14% (36)	11% (23)	−0.09
IV		3% (14)	4% (9)	2% (5)	−0.07
CCS: 0	391	26% (100)	27% (59)	24% (41)	−0.07
1		16% (63)	18% (40)	13% (23)	−0.13
2		25% (97)	25% (54)	25% (43)	0.01
3		16% (63)	14% (30)	19% (33)	0.15
4		17% (68)	16% (36)	19% (32)	0.06
EuroSCORE II	303	0.94/1.52/3.00	1.07/1.72/3.15	0.80/1.38/2.60	−0.20
GFR	439	56/74/90	53/69/90	58/75/90	0.17
Hemoglobin	454	12/13/15	12/13/15	12/14/15	0.08
LVEF: poor (<30%)	462	4% (17)	3% (8)	4% (9)	0.06
fair (30–50%)		36% (166)	41% (104)	30% (62)	−0.25
good (>50%)		60% (279)	56% (140)	66% (139)	0.22
Aspirin	471	79% (373)	80% (208)	79% (165)	−0.03
Thienidopiridine	471	54% (255)	52% (135)	57% (120)	0.11
Syntax score	188	18/23/29	19/25/30	17/22/27	−0.27
LAD	472	83% (394)	85% (220)	82% (174)	−0.09
LCX	475	51% (242)	56% (147)	44% (95)	−0.24
RCA	474	46% (218)	52% (135)	39% (83)	−0.26

**Table 2 ijerph-18-06694-t002:** Standardized mean differences (SMD), overlapping coefficient (OVL), and C-statistic computed the original dataset before the application of PS methods. For methods based on MI, the balance statistics are reported as median (minimum; maximum).

		Measure of Balance		
Missing Data	PS Estimation	SMD	OVL	C-Statistc
CC	LR	1.09	0.42	0.33
	GBM	2.44	0.73	0.46
	CBPS	1.07	0.38	0.29
MIND	LR	0.53	0.35	0.25
	GBM	2.11	0.69	0.44
	CBPS	0.53	0.33	0.24
SAEM	LR (SAEM)	0.7	0.28	0.19
GBM (surr.)	GBM	1.76	0.59	0.39
MI-AREGIMP	CBPS	0.7 (0.64;0.81)	0.27 (0.25;0.31)	0.19 (0.17;0.21)
	GBM	1.86 (1.56;2.3)	0.63 (0.54;0.74)	0.41 (0.37;0.45)
	LR	0.73 (0.68;0.84)	0.28 (0.26;0.3)	0.19 (0.18;0.22)
MI-BBPMM	CBPS	0.69 (0.65;0.75)	0.27 (0.25;0.29)	0.18 (0.17;0.2)
	GBM	1.99 (1.7;2.29)	0.67 (0.56;0.72)	0.43 (0.39;0.45)
	LR	0.72 (0.67;0.82)	0.27 (0.25;0.33)	0.19 (0.18;0.22)
MI-MICE	CBPS	0.74 (0.67;0.81)	0.28 (0.25;0.31)	0.2 (0.18;0.21)
	GBM	1.9 (1.69;2.31)	0.63 (0.58;0.74)	0.41 (0.39;0.46)
	LR	0.76 (0.71;0.84)	0.29 (0.27;0.32)	0.2 (0.19;0.22)

**Table 3 ijerph-18-06694-t003:** Measures of overall balance for sets of patients obtained after the application of PS methods, i.e., Standardized mean differences (SMD), overlapping coefficient (OVL), C-statistic, and proportion of retained individuals (PRI). For methods based on MI, the balance statistics are reported as median (minimum; maximum).

			PS Based Methods										
		NN				FM				PS-IPTW			
Missing Data	PS Estimation	SMD	OVL	C-Statistic	PRI	SMD	OVL	C-Statistic	PRI	SMD	OVL	C-Statistic	PRI
CC	*LR*	0.36	0.24	0.2	0.74	0.1	0.07	0.12	1	0.1	0.05	0.03	0.54
	*GBM*	0.16	0.09	0.1	0.21	0.44	0.26	0.32	1	2.19	0.66	0.44	0.97
	*CBPS*	0.27	0.13	0.12	0.74	0.19	0.15	0.13	1	0.28	0.22	0.06	0.8
MIND	*LR*	0.24	0.33	0.23	0.88	0.1	0.05	0.07	1	0.13	0.1	0.02	0.81
	*GBM*	0.2	0.13	0.11	0.32	0.32	0.22	0.22	1	1.79	0.58	0.39	0.95
	*CBPS*	0.25	0.31	0.23	0.88	0.1	0.06	0.1	1	0.2	0.11	0.04	0.85
SAEM	*LR (SAEM)*	0	0	0	0.42	0.01	0.01	0.06	1	0.03	0.1	0	0.88
GBM (surr.)	*GBM*	0	0	0	0.26	0.06	0.04	0.11	1	1.29	0.44	0.31	0.94
MI-AREGIMP	*CBPS*	0.06 (0.04;0.08)	0.04 (0.03;0.05)	0.02 (−0.01;0.03)	0.72 (0.69;0.75)	0.02 (0.01;0.04)	0.02 (0.01;0.03)	0.04 (0;0.09)	1	0.16 (0.1;0.19)	0.1 (0.06;0.13)	0.04 (0.02;0.05)	0.92 (0.9;0.93)
	*GBM*	0.14 (0.1;0.27)	0.1 (0.07;0.19)	0.07 (0.05;0.15)	0.39 (0.29;0.46)	0.12 (0.02;0.39)	0.1 (0.03;0.29)	0.12 (0.06;0.23)	1	1.5 (1.17;1.99)	0.5 (0.41;0.61)	0.34 (0.29;0.39)	0.95 (0.93;0.96)
	*LR*	0.08 (0.06;0.1)	0.05 (0.04;0.06)	0.03 (0.02;0.04)	0.72 (0.67;0.76)	0.02 (0.01;0.03)	0.02 (0.01;0.03)	0.03 (−0.01;0.08)	1	0.03 (0.01;0.06)	0.08 (0.03;0.11)	0 (0;0.01)	0.87 (0.84;0.89)
MI-BBPMM	*CBPS*	0.06 (0.04;0.08)	0.04 (0.03;0.05)	0.02 (0.01;0.03)	0.73 (0.7;0.76)	0.02 (0.01;0.03)	0.01 (0.01;0.02)	0.02 (−0.03;0.09)	1	0.14 (0.07;0.19)	0.08 (0.07;0.11)	0.04 (0.02;0.05)	0.93 (0.88;0.94)
	*GBM*	0.18 (0.11;0.22)	0.12 (0.06;0.16)	0.09 (0.05;0.13)	0.37 (0.29;0.46)	0.2 (0.06;0.41)	0.15 (0.05;0.28)	0.13 (0.08;0.29)	1	1.66 (1.3;1.99)	0.55 (0.41;0.62)	0.36 (0.31;0.4)	0.95 (0.94;0.96)
	*LR*	0.08 (0.06;0.1)	0.05 (0.04;0.07)	0.03 (0.02;0.04)	0.73 (0.68;0.76)	0.02 (0.01;0.03)	0.01 (0.01;0.02)	0.04 (−0.02;0.09)	1	0.03 (0.02;0.05)	0.08 (0.06;0.11)	0.01 (0;0.01)	0.88 (0.83;0.89)
MI-MICE	*CBPS*	0.07 (0.05;0.09)	0.04 (0.03;0.06)	0.03 (−0.02;0.03)	0.72 (0.68;0.74)	0.02 (0.01;0.03)	0.01 (0.01;0.02)	0.04 (−0.04;0.07)	1	0.14 (0.06;0.2)	0.08 (0.05;0.1)	0.04 (0.01;0.05)	0.91 (0.88;0.93)
	*GBM*	0.16 (0.12;0.26)	0.11 (0.08;0.19)	0.08 (0.06;0.15)	0.39 (0.29;0.46)	0.14 (0.05;0.24)	0.1 (0.04;0.18)	0.13 (0.08;0.19)	1	1.46 (1.3;2.05)	0.49 (0.44;0.64)	0.34 (0.31;0.41)	0.94 (0.93;0.95)
	*LR*	0.1 (0.08;0.11)	0.06 (0.04;0.07)	0.04 (0.03;0.04)	0.72 (0.67;0.74)	0.01 (0.01;0.02)	0.01 (0.01;0.02)	0.03 (−0.02;0.1)	1	0.02 (0;0.05)	0.07 (0.04;0.12)	0 (0;0.01)	0.85 (0.83;0.88)

## Data Availability

The data presented in this study are available on request from the corresponding author. The data are not publicly available due to privacy.

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
