# Peer review of "Propensity Score Analysis with Partially Observed Baseline Covariates: A Practical Comparison of Methods for Handling Missing Data"

_ijerph, 2021, doi:10.3390/ijerph18136694_

Round 1
Reviewer 1 Report
The paper submitted outlines the comparative strengths and weaknesses of a number of established statistical methodologies relating to propensity scoring methods used for handling missing baseline covariate data in patient datasets. Although essentially a comparison of statistical methodologies, the contextualisation of the study with respect to the analysis of data related to coronary disease will, in my opinion, make the paper of interest to the journal of the International Journal of Environmental Research and Public Health. As such this is an appropriate submission to the journal.
The introduction is well-written, concise and appropriately referenced. It contextualises the need for the study outlined in the paper and appropriately introduces the different scoring propensity methods that have been used in similar studies to account for missing baseline data.
The methods section is somewhat long, and I have recommended (see below) that some material relating to the introduction and contextualisation of propensity scoring methods be relocated to the introduction section. The methodology that is presented seems robust and scientifically appropriate.
The results are generally clearly laid out and unambiguous in presentation.
I am not sure the first paragraph of the discussion adds to the paper in that it is simply a summary of what has preceded (not actual discussion) and the paper is comprehensively written as not to need this.
The discussion points are directly related to the experiments conducted and appropriately critically contextualised. The limitations of the study are well recognised and clearly presented. The conclusions are reasonable and drawn appropriately from the experiments conducted.
As such, I would recommend the paper for publication subsequent to the author’s consideration of the suggested minor amendments given below.
Minor Amendments
Page 3 – 148/149. Tautology needs removing (Previous studies previous papers…)
The structure of the paper is giving me some cause for concern as it appears that methodology, introduction and results are not particularly well delineated. Specifically
- I would argue that sections 2.1 and 2.2 are actually better suited to introduction (they comprise an introduction to the different types of propensity scoring and critical analysis of comparative strengths and weaknesses). I would encourage the authors to consider relocating these sections to the introduction part of the manuscript.
- There is a little bit of confusion between the past and present tense in the methodology section which makes the manuscript more difficult to read than it could be. Could the authors consider reviewing to make everything past tense?
- The ‘results’ for the decision tree exploration are actually a combination of methods and results. Would it not be best to detail the methods for the decision tree analysis in the methods section?
Explicitly labelling the axes on the Figure 3 Correlogram would help to aid understanding of this image
Author Response
Open Review
(x) I would not like to sign my review report
( ) I would like to sign my review report
English language and style
( ) Extensive editing of English language and style required
(x) Moderate English changes required
( ) English language and style are fine/minor spell check required
( ) I don't feel qualified to judge about the English language and style
|
Yes |
Can be improved |
Must be improved |
Not applicable |
|
|
Does the introduction provide sufficient background and include all relevant references? |
( ) |
(x) |
( ) |
( ) |
|
Is the research design appropriate? |
(x) |
( ) |
( ) |
( ) |
|
Are the methods adequately described? |
(x) |
( ) |
( ) |
( ) |
|
Are the results clearly presented? |
( ) |
(x) |
( ) |
( ) |
|
Are the conclusions supported by the results? |
(x) |
( ) |
( ) |
( ) |
Comments and Suggestions for Authors
The paper submitted outlines the comparative strengths and weaknesses of a number of established statistical methodologies relating to propensity scoring methods used for handling missing baseline covariate data in patient datasets. Although essentially a comparison of statistical methodologies, the contextualisation of the study with respect to the analysis of data related to coronary disease will, in my opinion, make the paper of interest to the journal of the International Journal of Environmental Research and Public Health. As such this is an appropriate submission to the journal.
The introduction is well-written, concise and appropriately referenced. It contextualises the need for the study outlined in the paper and appropriately introduces the different scoring propensity methods that have been used in similar studies to account for missing baseline data.
The methods section is somewhat long, and I have recommended (see below) that some material relating to the introduction and contextualisation of propensity scoring methods be relocated to the introduction section. The methodology that is presented seems robust and scientifically appropriate.
The results are generally clearly laid out and unambiguous in presentation.
I am not sure the first paragraph of the discussion adds to the paper in that it is simply a summary of what has preceded (not actual discussion) and the paper is comprehensively written as not to need this.
We would like to thank the reviewer for the comment. We removed the first paragraph of the discussion.
The discussion points are directly related to the experiments conducted and appropriately critically contextualised. The limitations of the study are well recognised and clearly presented. The conclusions are reasonable and drawn appropriately from the experiments conducted.
As such, I would recommend the paper for publication subsequent to the author’s consideration of the suggested minor amendments given below.
We would like to thank the reviewer for the helpful feedback.
Minor Amendments
Page 3 – 148/149. Tautology needs removing (Previous studies previous papers…)
We thank the reviewer for pointing this out. We modified the sentence accordingly.
The structure of the paper is giving me some cause for concern as it appears that methodology, introduction and results are not particularly well delineated. Specifically
- I would argue that sections 2.1 and 2.2 are actually better suited to introduction (they comprise an introduction to the different types of propensity scoring and critical analysis of comparative strengths and weaknesses). I would encourage the authors to consider relocating these sections to the introduction part of the manuscript.
We thank the reviewer for the suggestion. We moved sections 2.1 and 2.2 in the introduction part.
- There is a little bit of confusion between the past and present tense in the methodology section which makes the manuscript more difficult to read than it could be. Could the authors consider reviewing to make everything past tense?
We thank the reviewer for the comment. We modified the motivating example and statistical analysis sections accordingly. However, we preferred to keep the present tense for the sections that describe the PS framework and methods for missing data, as it is common practice for the studies from the methodological literature that introduce or describe a statistical method.
- The ‘results’ for the decision tree exploration are actually a combination of methods and results. Would it not be best to detail the methods for the decision tree analysis in the methods section?
We thank the reviewer for the comment. We moved the description of the decision tree method to the methodological section.
Explicitly labelling the axes on the Figure 3 Correlogram would help to aid understanding of this image
We thank the reviewer for the suggestion. We modified Figure 3 accordingly.
Reviewer 2 Report
In general, the article is rather well-written and it can be interesting for the readers.
However, the paper focuses on the comparison of the existing methods using one (and still undergoing) case study.
There is no introduction of the new methods or new applications of the previous ones.
Therefore, I suggest providing some additional details concerning the “most promising methods” indicated by the authors (i.e. PS with the SAEM method together with FM). There are a few sentences that should be checked (like “Previous studies previous paper have shown”. p. 3).
Author Response
Open Review
(x) I would not like to sign my review report
( ) I would like to sign my review report
English language and style
( ) Extensive editing of English language and style required
( ) Moderate English changes required
(x) English language and style are fine/minor spell check required
( ) I don't feel qualified to judge about the English language and style
|
Yes |
Can be improved |
Must be improved |
Not applicable |
|
|
Does the introduction provide sufficient background and include all relevant references? |
( ) |
(x) |
( ) |
( ) |
|
Is the research design appropriate? |
(x) |
( ) |
( ) |
( ) |
|
Are the methods adequately described? |
( ) |
(x) |
( ) |
( ) |
|
Are the results clearly presented? |
(x) |
( ) |
( ) |
( ) |
|
Are the conclusions supported by the results? |
(x) |
( ) |
( ) |
( ) |
Comments and Suggestions for Authors
In general, the article is rather well-written and it can be interesting for the readers.
However, the paper focuses on the comparison of the existing methods using one (and still undergoing) case study.
There is no introduction of the new methods or new applications of the previous ones.
Therefore, I suggest providing some additional details concerning the “most promising methods” indicated by the authors (i.e. PS with the SAEM method together with FM). There are a few sentences that should be checked (like “Previous studies previous paper have shown”. p. 3).
We thank the reviewer for the precious comments and suggestions. We provided further details in the methods, discussion, and conclusion sections on PS with SAEM methods and FM method, highlighting findings from a recent study and contextualizing previous results with the findings of our study. Moreover, we corrected the sentence on page 3 accordingly.